# Supramolecular Dimer as High-Performance pH Probe: Study on the Fluorescence Properties of Halogenated Ligands in Rigid Schiff Base Complex

**DOI:** 10.3390/ijms24119480

**Published:** 2023-05-30

**Authors:** Jiajun Xu, Meifen Huang, Liang Jiao, Haijun Pang, Xia Wang, Rui Duan, Qiong Wu

**Affiliations:** 1Department of Chemical Science and Technology, Kunming University, Kunming 650214, China; xjj1999122@gmail.com (J.X.); 18213054216@163.com (X.W.); dr57512023@163.com (R.D.); 2College of Physics Science and Technology, Kunming University, Kunming 650214, China; hmf668648@gmail.com (M.H.); kysy3510@gmail.com (L.J.); 3The School of Material Science and Chemical Engineering, Harbin University of Science and Technology, Harbin 150040, China; panghj116@163.com; 4Yunnan Key Laboratory of Metal-Organic Molecular Materials and Device, School of Chemistry and Chemical Engineering, Kunming University, Kunming 650214, China

**Keywords:** Schiff base complex, fluorescent probe, pH detection, halogen bond, supramolecular aggregates, DFT studies

## Abstract

The development of high-performance fluorescence probes has been an active area of research. In the present work, two new pH sensors *Zn-3,5-Cl-saldmpn* and *Zn-3,5-Br-saldmpn* based on a halogenated Schiff ligand (*3,5-Cl-saldmpn* = N, N′-(3,3′-dipropyhnethylamine) bis (3,5-chlorosalicylidene)) with linearity and a high signal-to-noise ratio were developed. Analyses revealed an exponential intensification in their fluorescence emission and a discernible chromatic shift upon pH increase from 5.0 to 7.0. The sensors could retain over 95% of their initial signal amplitude after 20 operational cycles, demonstrating excellent stability and reversibility. To elucidate their unique fluorescence response, a non-halogenated analog was introduced for comparison. The structural and optical characterization suggested that the introduction of halogen atoms can create additional interaction pathways between adjacent molecules and enhance the strength of the interaction, which not only improves the signal-to-noise ratio but also forms a long-range interaction process in the formation of the aggregation state, thus enhancing the response range. Meanwhile, the above proposed mechanism was also verified by theoretical calculations.

## 1. Introduction

pH plays a significant role in biological and physiological processes as it influences the function of enzymes and other biomolecules that are vital for various cellular activities such as energy generation, DNA replication, and signaling cascades [1,2,3,4]. Minor fluctuations in the pH environment can impact the function of intracellular enzymes and other biomolecules, altering metabolic reactions and other critical cellular processes [5,6]. pH also governs the electrical properties of cell membranes, influencing ion transport across them [7]. Furthermore, pH changes can trigger signaling pathways in cells, leading to variations in numerous physiological processes [8]. Therefore, pH is an essential factor in diverse biological systems, and accurate pH measurement is necessary to gain deeper insights into these processes. Given that most physiological and pathological processes occur within the pH range of 5.0 to 7.0, developing a real-time and high-precision pH detection method in this range is of profound importance [1,3].

pH test paper offers a simple and convenient way for semi-quantitative pH measurement via visual color changes, among which fluorescence detection technology is widely investigated and applied due to its high sensitivity, good selectivity and the ability to meet the demand of special detection conditions [2,9,10]. So far, fluorescent probes based on intramolecular charge transfer (ICT), fluorescence resonance energy transfer (FRET), photo-induced electron transfer (PET) and other mechanisms have been widely developed and used for pH detection [11,12,13]. However, we have noticed that the formation mechanism of the fluorescence signal is mainly based on changes in electronic or covalent structures, which can only generate a relatively weak response in terms of changes in external pH environments. Furthermore, the regulation strategy of traditional probes mainly focuses on the redesignation of the framework of the probes. Nevertheless, alterations to the molecular framework may impede the linear control of fluorescence demeanor or even fundamentally change the fluorescence behaviors of the probes. Therefore, the design and synthesis of high-performance probes with controllable fluorescence behavior to meet the specific demands of complex applications remains a challenge.

Supramolecular aggregates are formed through intermolecular recognition and their physicochemical properties can be adjusted based on the degree of aggregation without changing their internal molecular structure [14,15]. The driving forces behind supramolecular assembly can typically be attributed to intermolecular interactions such as π–π stacking, hydrogen bonding, halogen bonding, and electrostatic interactions, all of which can be significantly influenced by the changes in the H^+^ environment [15,16,17,18,19]. Halogen atoms can affect the electronic structure of the benzene ring through π–π conjugation or induction effects, and the substitution of hydrogen atoms on the benzene ring with different halogen atoms can control the surface charge distribution and regulate the intermolecular interactions [20,21,22]. Additionally, the substitutional halogen atoms also form extra charge transfer paths between fluorescent molecules, resulting in a more concentrated charge distribution and weaker electronic coupling between the molecules, which accelerates the rates of both radiative and non-radiative transitions [23,24,25]. More importantly, the changes in halogen atoms do not affect the mode of intermolecular interactions, enabling the theoretical feasibility of enhancing the signal-to-noise ratio of the probe by regulating the intermolecular interactions’ strength through different halogen atoms.

Salen-type molecules represent a class of Schiff base compounds formed by the condensation reactions of different salicylaldehydes and diamines, resulting in diverse structures and properties [15,20,26]. As an important molecule in materials science, salen has been extensively studied for decades, and its specific fluorescence behaviors have been widely applied in detection domains, such as biological imaging, metal ion recognition, explosive detection, and environmental monitoring [27,28,29,30]. Nonetheless, the confined fluorescence intensity and inferior stability of salen compounds significantly hamper their utilization as high-performance pH sensors.

In our previous work, we found that the pentadentate salen-type ligand exhibited enhanced stability upon halogenation, which could serve as a high-performance supramolecular fluorescent probe to detect pH between 5.0 and 6.0 [15]. However, since most physiological and pathological processes occur between 5.0–7.0, such a detection range is unable to satisfy the requirements of practical applications; further broadening the detection range and enabling them to more closely cover the full physiological pH range is of great importance [1,3]. Based on the aforementioned considerations, we designed and synthesized two new fluorescent probes based on rigid, halogenated Schiff base complexes, *Zn-3,5-Cl-saldmpn* and *Zn-3,5-Br-saldmpn*, and introduced a non-halogenated analog (*Zn-saldmpn*) for comparison. The results indicated that the signal-to-noise ratio of the halogenated Schiff base probes was substantially enhanced, while the fluorescence intensities of *Zn-3,5-Cl-saldmpn* and *Zn-3,5-Br-saldmpn* increased by 12.2 and 6.7 times, respectively. Additionally, the halogen-regulated Schiff base complexes exhibited outstanding stability, selectivity, and reversibility. The study also performed an in-depth analysis of the single-crystal structure of the compounds and the intermolecular interaction between adjacent molecules. Theoretical calculations were also performed to investigate the potential mechanisms underlying the varying signal-to-noise ratios caused by the substitution of different atoms.

## 2. Results

### 2.1. Synthesis and Characterization

#### 2.1.1. Synthesis of Zn-3,5-Cl-Saldmpn

The synthetic route of *Zn-3,5-Cl-saldmpn* is shown in Figure 1 (R_1_). First, 0.191 g (1.0 mmol) of 3,5-dichlorosalicylaldehyde and 82 μL (0.5 mmol) of N′, N-bis(3-aminopropyl) methylamine were added to 50 mL of ethanol and stirred at room temperature for 2 h. Subsequently, 0.069 g (0.5 mmol) of ZnCl_2_ were added to the mixture, and the resulting solution turned yellow. The solution was then stirred for 1 h and filtered. After being left at room temperature for three days, clear yellow crystals of *Zn-3,5-Cl-saldmpn* were obtained, and the crystalline product was washed with ethanol and dried in air (yield 72.6%). FT-IR (KBr, cm^−1^): 1631 (C=N), 1450 (CH_3_), 1162 (C–O), 864 (Zn–N), 758 (C–Cl), and 557 (Zn–O) (Appendix A).

In the UV absorption spectrum, the ligand H_2_-3,5-Cl-saldmpn showed two strong absorption peaks located at 282 nm and 328 nm, respectively. These two peaks were attributed to the π–π* transition of the benzene ring in the ligand molecule. In addition, a similar strong absorption peak was observed at 420 nm, which was attributed to the n–π* transition absorption peak on the C=N bond in the ligand molecule. Compared with the ligand, the position of the UV absorption peak of the metal complex *Zn-3,5-Cl-saldmpn* changed or disappeared. The original π–π* transition absorption peaks corresponding to the benzene ring at 282 nm and 328 nm of the ligand were red-shifted to 290 nm and 375 nm, respectively, in the complex. This indicated that the ligand H_2_-3,5-Cl-saldmpn had coordinated with the Zn(II) ions. At the same time, the absorption peak of the ligand H_2_-3,5-Cl-saldmpn near 420 nm disappeared in the complex. This indicated that the oxime nitrogen atom had also coordinated with the metal Zn(II) ions.

#### 2.1.2. Synthesis of Zn-3,5-Br-Saldmpn

The synthetic route of *Zn-3,5-Br-saldmpn* is shown in Figure 1 (R_2_), in which the procedures were the same as for *Zn-3,5-Cl-saldmpn*, except that 3,5-dibromosalicylaldehyde (0.280 g, 1.0 mmol) was used instead (yield: 71.8%). FT-IR (KBr, cm^−1^): 1631 (C=N), 1446 (CH_3_), 1149 (C–O), 871 (Zn–N), 702 (C–Br), and 553 (Zn–O) (Appendix A).

In the UV absorption spectrum, the ligand H_2_-3,5-Br-saldmpn showed two strong absorption peaks located at 286 nm and 332 nm, respectively. These two peaks were attributed to the π–π* transition of the benzene ring in the ligand molecule. In addition, a similar strong absorption peak was observed at 422 nm, which was attributed to the n–π* transition absorption peak on the C=N bond in the ligand molecule. Compared with the ligand, the position of the UV absorption peak of the metal complex *Zn-3,5-Br-saldmpn* changed or disappeared. The original π–π* transition absorption peaks corresponding to the benzene ring at 286 nm and 332 nm of the ligand were red-shifted to 297 nm and 383 nm, respectively, in the complex. This indicated that the ligand H_2_-3,5-Br-saldmpn had coordinated with the Zn(II) ions. At the same time, the absorption peak of the ligand H_2_-3,5-B-saldmpn near 420 nm disappeared in the complex. This indicated that the oxime nitrogen atom had also coordinated with the metal Zn(II) ions.

#### 2.1.3. Synthesis of Zn-Saldmpn

The synthetic route of *Zn-saldmpn* is shown in Figure 1 (R_3_), in which the procedures were the same as *Zn-3,5-Cl-saldmpn* except that salicylaldehyde (0.125 g, 1.0 mmol) was used instead (yield: 64.5%). FT-IR (KBr, cm^−1^): 1627 (C=N), 1473 (CH_3_), 1153 (C–O), 899 (Zn–N), 758 (C–H), and 597 (Zn–O) (Appendix A).

In the UV absorption spectrum, the ligand H_2_-saldmpn showed a strong absorption peak at 254 nm and a broader peak at 316 nm, which were attributed to the π–π* transition of the benzene ring and n–π* transitions of the C=N bonds, respectively. Compared with the ligand H_2_-saldmpn, the absorption peak of *Zn-saldmpn* was red-shifted to different degrees. The characteristic absorption band of the π–π* transition of the benzene ring was red-shifted to around 260 nm, which may have been due to the coordination of the O atom on the phenolic hydroxyl group, while the characteristic absorption band of the n–π* transition of the C=N group was red-shifted to 350 nm (Appendix A).

CCDC 2254822, 2254823 and 680620 [31] contain the supplementary crystallographic data for this paper. These data can be obtained free of charge from The Cambridge Crystallographic Data Centre via www.ccdc.cam.ac.uk/data_request/cif (accessed on 6 May 2023).

### 2.2. Photophysical Properties

To explore the optical responses of the two halogenated Schiff base complexes to pH, we first determined their fluorescence intensity changes at different pH values (Appendix A). The experiment showed that as the pH transitioned from 2.0 to 5.0, the fluorescence intensity remained largely invariant, and overall exhibited fluorescence quenching effects. In contrast, as the pH ascended from 7.0 to 12.0, the fluorescence emission intensified but without marked variations. Strikingly, within the pH window spanning 5.0 to 7.0, the fluorescence intensities of both the halogenated Schiff base complexes exhibited a dramatic intensification (Figure 1a,b). The fluorescence intensities of *Zn-3,5-Cl-saldmpn* and *Zn-3,5-Br-saldmpn* were amplified by 12.2 and 6.7 times, respectively (Appendix A), and both showed excellent linearity within this range (Figure 1c,d), with R^2^ values of 0.996 and 0.990, respectively. Employing the Henderson–Hasselbalch equation [32,33], the pKa values of *Zn-3,5-Cl-saldmpn* and *Zn-3,5-Br-saldmpn* were calculated to be 6.05 and 6.11, respectively (Appendix A). In addition, we observed the change in fluorescence intensity from pH 5.0 to 7.0 under a 365 nm UV lamp (Figure 1e,f), which was consistent with the FL results, and the associated changes were discernible via visual examination. As is widely recognized, coordination complexes have stability within a certain pH range [15,34], particularly proximal to the pH of their synthesis conditions (*Zn-3,5-Cl-saldmpn*: pH = 5.8; *Zn-3,5-Br-saldmpn*: pH = 5.7). However, observing fluorescence behavior with an exceptionally robust signal-to-noise ratio near their synthesis pH is rather uncommon. To gain a better comprehension of this unusual phenomenon, we introduced the non-halogenated Schiff base complex *Zn-saldmpn* (Appendix A) for comparison. Although the fluorescence intensity of *Zn-saldmpn* also increased as the pH changed from 5.0 to 7.0, its signal-to-noise ratio remained marginal (only 1.7).

This could be attributed to the decline in solution pH eliciting an elevation in solvent polarity, as well as the hydrogen atoms covering the non-halogenated isomer having a stronger affinity for polar solvents. Upon reaching the critical point, a relatively minor polarity change can cause the probe molecules to transform from free molecules to an aggregation state. However, for halogenated species, the incorporation of halogen atoms can augment molecular hydrophobicity and attenuate their response to polar solvents [35], resulting in a significant increase in the fluorescence response range of polar solvents. The following computational analysis provides further insight into this phenomenon.

The UV absorption spectra of three Schiff base derivatives are shown in Figure 2. As the pH descended from 7.0 to 5.0, the absorption peak of *Zn-3,5-Cl-saldmpn* at 380 nm gradually decreased, concomitant with the emergence of a novel absorption peak at 345 nm. Similar to *Zn-3,5-Cl-saldmpn*, the disappeared and newly formed absorption peaks of *Zn-3,5-Br-saldmpn* occurred at 383 nm and 341 nm, respectively. In contrast, the absorption peak of *Zn-saldmpn* at 352 nm lowered with the decrease in pH, and no new absorption peak was observed. Moreover, as the concentration of H^+^ increased, the color change from colorless to yellow of *Zn-3,5-Cl-saldmpn* and *Zn-3,5-Br-saldmpn* could be visually discerned under ambient light, while no such phenomenon occurred for *Zn-saldmpn* (Figure 2 inset). Therefore, the halogen-regulated Schiff base probes can serve as “naked-eye” colorimetric sensors for acidic pH values.

### 2.3. Sensing Mechanism

To investigate the anomalous optical behavior of the halogenated Schiff base complexes, we assessed and compared the quantum yields and fluorescence lifetimes of the three Schiff base complex derivatives. As delineated in Appendix A, the quantum yield of *Zn-3,5-Cl-saldmpn* increased substantially from 4.4% to 19.4% as the pH changed from 5.0 to 7.0. Similar to *Zn-3,5-Cl-saldmpn*, the quantum yield of *Zn-3,5-Br-saldmpn* also increased by 2.7% (Appendix A), whereas the quantum yield of *Zn-saldmpn* only showed an increasing trend over a small pH range from 5.0 to 6.0 (Appendix A). In terms of fluorescence lifetimes (Appendix A), the fluorescence lifetimes of *Zn-3,5-Cl-saldmpn* and *Zn-3,5-Br-saldmpn* also increased with the change in pH values, increasing by 4.03 ns and 0.16 ns, respectively. However, the variations in *Zn-saldmpn* were trivial and virtually negligible.

We then calculated the radiative decay rate constants (k_r_) and non-radiative decay rate constants (k_nr_) for the three Schiff base complexes using quantum yields and fluorescence lifetimes (Table 1) [36]. It can be seen that as the pH increased from 5.0 to 7.0, the k_r_ values of *Zn-3,5-Cl-saldmpn* and *Zn-3,5-Br-saldmpn* both increased, while the k_nr_ values decreased, which implied that the transmission efficiency of the emitted light was enhanced because the energy of the excitation light was more easily dispersed into radiative energy [37]. It should be noted that under the same pH conditions, the k_nr_ value of *Zn-3,5-Br-saldmpn* was much larger than that of *Zn-3,5-Cl-saldmpn*, which may have been caused by the heavy atom effect [38]. Specifically, the heavy atom effect can cause the compression of electron orbitals as well as the movement of energy levels, resulting in an acceleration of the average speed of electrons and an enhancement of the strength of the interaction between electrons and the lattice [39,40]. Therefore, although the substitution of the bromine atom results in the formation of stronger intermolecular interactions, the presence of the heavy atom effect leads to *Zn-3,5-Br-saldmpn* not yielding higher quantum yields.

The enhancement of the interaction increased the rate constant of non-radiative transitions, resulting in fluorescence quenching or reduction, which reduced the signal-to-noise ratio of *Zn-3,5-Br-saldmpn* at different pHs to a certain extent. Interestingly, when the pH was increased from 5.0 to 6.0, the k_r_ value of *Zn-saldmpn* showed a certain degree of increase, indicating that the transmission of the emitted light became easier, which is consistent with the previously observed fluorescence enhancement phenomenon. However, as the pH continued to increase to 7.0, the k_r_ value of *Zn-saldmpn* showed a downward trend. This complex change greatly limited the signal-to-noise ratio of *Zn-saldmpn*. In addition, the k_nr_ values of the three Schiff base complex derivatives were much higher than the k_r_, indicating the crucial role of intermolecular interactions in their fluorescence behavior. 

Relevant research as well as our recent work progress indicate that H^+^ as a driving force can effectively regulate the strength of intermolecular interactions and construct specific supramolecular aggregates [15,20]. Since intermolecular interactions inhibit the emission process, the radiative decay rate constant k_r_ can be significantly reduced with the formation of aggregate states. Furthermore, according to the exciton theory, fluorescent molecules can be seen as point dipoles, and the excited states of aggregates are split into two energy levels through the interaction of transition dipoles [41]. As the dipoles are parallel to each other, the repulsive force can cause a higher energy state, resulting in a blue shift in the UV absorption spectrum, which is consistent with the previous observations mentioned in the text [41]. Based on these observations, it is plausible to suggest that the difference in signal-to-noise ratios observed in the fluorescence signals of the Schiff base complex derivatives may be related to the formation of aggregates.

While investigating the influence of aggregate states on the fluorescence intensity of the studied compounds, as shown in Figure 3a, although the crystals of these three Schiff base complex derivatives exhibited a yellow color under sunlight, they all showed fluorescence quenching under the 365 nm UV lamp due to the formation of molecular aggregates and non-radiative transitions, which is consistent with the previously mentioned increase in the non-radiative decay rate constant. This is a typical aggregation-caused quenching (ACQ) behavior (Figure 3b) [42,43].

With the goal of studying the influence of the aggregation degree on the fluorescence signal-to-noise ratio of Schiff base complexes, we performed dynamic light scattering (DLS) experiments and observed the changes in the Tyndall phenomenon for the three Schiff base complexes at different pHs. As the pH further decreased, the particle size scales of *Zn-3,5-Cl-saldmpn* and *Zn-3,5-Br-saldmpn* increased significantly (Figure 3c and Appendix A), reaching an impressive 292.4 nm and 262.4 nm, respectively. In contrast, the particle size of the conventional Schiff base complex *Zn-saldmpn* remained almost unchanged during the decrease in pH from 7.0 to 6.0; when the pH was reduced to 5.0, the particle size of *Zn-saldmpn* rapidly increased to 111.6 nm (Figure 3d). Meanwhile, during the process of the pH decreasing from 7.0 to 5.0, the Tyndall effect of the halogenated Schiff base complexes showed a significant enhancement trend (Appendix A), and the brightest light column appeared at pH 5.0. However, for the non-halogenated analog *Zn-saldmpn*, only a faint Tyndall beam appeared at pH 5.0 (Appendix A). Since the Tyndall phenomenon can well reflect the size of nanoparticles in solution, and because the brightness of the light beam of the Tyndall effect is positively correlated with the size of the solute particles in the solution [44,45], the above results are completely consistent with the DLS data.

At the same time, we used scanning electron microscopy (SEM) to observe the self-assembly pathways of the three Schiff base complexes at the microstructure level. At pH 7.0, the three Schiff bases exhibited a dispersed structure (Figure 3e and Appendix A). However, when the pH lowered to 6.0, the halogenated Schiff bases exhibited an aggregation trend and formed small particles. As the pH continued to decrease to 5.0, *Zn-3,5-Cl-saldmpn* and *Zn-3,5-Br-saldmpn* formed a cotton-like aggregation state. In contrast, *Zn-saldmpn* underwent almost no morphology changes as the pH decreased from 7.0 to 5.0, and only appeared as a block-like aggregate at pHs down to 5.0 (Figure 3f).

The aforementioned results indicate that the substitution of halogen atoms can effectively enhance the aggregation degree of probe molecules. Therefore, we speculate that the mechanism is mainly reflected in the following two aspects: First, compared with halogen bonds, the hydrogen bond is relatively more directional. This is mainly due to the large charge difference between the hydrogen atom and the electronegative atom, which makes the angle of D-H… A (donor atom (D), the hydrogen atom (H) and the acceptor atom (A)) more fixed. Although the hydrogen bond energy is large, its formation conditions are relatively strict, resulting in fewer hydrogen bonds and a relatively lower interaction energy.

The strong polarizing effect of halogen atoms leads to the formation of multiple uneven charge distribution regions both in the halogen atom as well as the bonded atoms. For example, the halogen atom attracts more electron density from the bonded atoms to form a higher negative charge density region that is perpendicular to the covalent bond, while the polarization of the electron cloud leads to the appearance of a positively charged region (σ-hole) along the bonding direction, which leads to a slightly lower bonding energy for halogen bonds but creates more pathways for interaction with surrounding molecules, resulting in overall higher intermolecular forces. For molecules dominated by H-aggregation, stronger intermolecular interactions will cause an increase in the rate of non-radiative transitions, making the fluorescence quenching phenomenon more pronounced, which is the main reason for the significantly high signal-to-noise ratio of *Zn-3,5-Cl-saldmpn* and *Zn-3,5-Br-saldmpn*. However, the heavy atom effect is significant in Br-substituted analogous, causing the non-radiative transition rate constant of *Zn-3,5-Br-saldmpn* to be much larger than that of *Zn-3,5-Cl-saldmpn*.

### 2.4. Single-Crystal X-ray and Theoretical Analysis

Although the fluorescence behavior of the reported compounds comes from the intermolecular interactions in the solution environment, the solid-state stacking mode is crucial to understanding the profound mechanism [46,47]. Therefore, to obtain single-crystal structural information, we reacted the halogenated ligand H_2_-saldmpn (H_2_-χ-saldmpn) with zinc chloride in a 1:1 stoichiometric ratio in anhydrous ethanol. A colorless crystal suitable for single-crystal diffraction was obtained after one week (Appendix A).

The X-ray structural analysis showed that both crystallized in the monoclinic system (space group P21/c). Unexpectedly, the asymmetric unit incorporated two crystallographically independent, neutral Zn-χ-saldmpn units (abbreviated as [Zn1] and [Zn2]) (Figure 4b). According to the search results from the Cambridge Crystallographic Data Centre (CSD), although some non-halogenated saldmpn derivatives can be found in the database, all the deposited structures exist as single molecules. In addition to a halogenated trimer we previously reported [15], the Zn-χ-saldmpn series compounds (*Zn-3,5-Cl-saldmpn* and *Zn-3,5-Br-saldmpn*) are the second supramolecular compounds based on saldmpn ligands.

As shown in Figure 4a and Appendix A, each saldmpn^2−^ served as a pentadentate ligand, providing an N_3_O_2_ coordination sphere for the central atom, comprising one amine (Nam), two imine nitrogen atoms (Nim), and two phenolic oxygen atoms O*. The longest bond lengths around the metal center were the Zn-Nam bonds, ranging from 1.960 to 2.196 Å. The Zn-Nam bonds were the longest (2.149–2.196 Å), the Zn-Nim bonds were marginally abbreviated compared to the Zn-Nam bonds (2.049–2.087 Å), and the Zn-O* bonds were curtailed (1.960–2.00 Å). The lengths of these characteristic bonds were in the normal range of previously documented saldmpn-type zinc complexes. It is worth noting that due to the central atom connecting with the saldmpn^2−^ ligand in a pentacoordinate mode, there are two possible geometric shapes: square pyramidal (SP) and trigonal bipyramidal (TB). These two coordination geometries can be well determined by the Addison parameter τ [48]. From the calculated results, we can see that the τ values of the four zinc atoms in the two compounds showed some differences. The values of Zn1 and Zn2 in *Zn-3,5-Cl-saldmpn* were τ1 = 0.738 and τ2 = 0.828, while the τ values of Zn1 and Zn2 in *Zn-3,5-Br-saldmpn* were τ1 = 0.7165 and τ2 = 0.805, which indicated that the geometric shapes of the two zinc atoms were identical and belonged to the SP configuration.

The analysis of the weak intramolecular and intermolecular interactions revealed that in addition to classical hydrogen bonding, other types of intermolecular interactions were formed in the supramolecular dimer structures, including halogen bonds and π–π stacking. The interactions between the two molecules can be divided into three parts from left to right: the first part involved stronger bifurcated C–H...O interactions (C27-H27...O2, C28-H28...O2), the second part included π–π interactions [Cg1..Cg2] ((Cg1: C1-C2-C3-C4-C5-C6; (Cg2: C1-C2-C3-C4-C5-C6)) located in the middle of the supermolecule, while the third part was dominated by halogen bond C–H...Cl and C–H...O interactions. Under the synergistic effect of these intermolecular interactions, these two neutral molecules were interconnected to each other, forming a unique dimeric supramolecular structure. Further, as shown in Figure 4c–e, C–H...Cl interactions connected adjacent dimeric supramolecules into one-dimensional chains.

In order to better comprehend the chemical reactivity, electronic structure, and optical properties of the compounds reported herein, we conducted theoretical calculations using density functional theory (DFT) at the B3LYP functional level with the 6-311 + G(d,p) basis set [49]. It should be noted that experimental results showed that the fluorescence intensity at pH = 7.0 mainly originated from the monomer, so in this work, the atomic coordinates for optimization originated from the crystal structure of the Zn1 fragment. In addition, to shed light on the influence of halogenated ligands on the fluorescence behavior, we also used the non-halogenated isomer *Zn-saldmpn* for comparison. As shown in Appendix A, the optimized geometry of the Zn1 fragment was in good agreement with the crystal data, suggesting the good stability and rigidity of the reported compound.

The electron density distribution of Zn-χ-saldmpn in the HOMO orbital was distributed throughout the halogenated benzene rings, while the LUMO electron was mainly concentrated in the amine region. Ascribed to the introduction of halogen atoms, the electron distribution area of the benzene rings increased, indicating an improvement in the molecular conjugation, causing the neighboring molecules to be prone to forming H-aggregates via π–π interactions [50,51]. In comparison, the conjugated plane of the non-halogenated isomer was smaller, and the rigidity of the structure resulted in more apparent steric effects, which led to a decrease in the strength of the interaction with the surrounding molecules. On the other hand, from the numerical perspective, compared with *Zn-saldmpn*, the LUMO energy levels of *Zn-3,5-Cl-saldmpn* and *Zn-3,5-Br-saldmpn* were reduced from −1.4376 to −2.143 and −1.885 eV, respectively, indicating that the substitution of halogen atoms also favored the electron-accepting ability of Saldmpn-type complexes. Additionally, the energy gap of *Zn-3,5-Cl-saldmpn* and *Zn-3,5-Br-saldmpn* was smaller than that of *Zn-saldmpn* by 0.110 and 0.134 eV, respectively (Figure 5a–c). Regarding the energy perspective, a lower band gap facilitated the electron-extracting process, which indicated that the halogenated Zn-χ-saldmpn complexes had a higher chemical reactivity than that of the non-halogenated derivatives.

To more accurately describe the intensity of the interactions between supramolecular dimers, we calculated the interaction energies of the dimers and introduced a non-halogenated isomer for comparison (Figure 5d–f). The results show that the energies of the three compounds were ΔE_1_ = −93.16 kJ/mol (*Zn-3,5-Cl-saldmpn*), ΔE_2_ = −100.56 kJ/mol, and ΔE_3_ = −56.15 kJ/mol, revealing that the dimer energy of the halogenated compounds was much higher than that of the non-halogenated derivative. These results suggested that halogen substitution not only changed the arrangement of molecules but also increased the interaction energy between neighboring molecules. Meanwhile, combined with the experimental results, the higher interaction energy implied a stronger and “long-range” interaction process between the probe molecules, which not only increased the degree of molecular aggregation but also resulted in a broader response range of the detection environment.

### 2.5. Stability, Anti-Interference and Reversibility

Stability is a key indicator for probes in practical applications [52]. Therefore, we evaluated the stability of the three Schiff base complexes by measuring the fluorescence intensity at different time intervals. The fluorescence intensity of *Zn-3,5-Cl-saldmpn* and *Zn-3,5-Br-saldmpn* remained stable and unchanged over 15 days under pH values of 7.0 and 5.0 (Figure 6), suggesting a good stability of the reported halogenated compounds. However, the comparison result of *Zn-saldmpn* was unsatisfactory (Figure 6c). Under the pH 7.0 condition, the intensity began to decrease after 4 days; in the acidic environment (pH = 5.0), an obvious fluctuation in fluorescence intensity occurred after only 1 day. After 7 days, its fluorescence intensity at pH values of 7.0 and 5.0 almost overlapped, and visible precipitation formed in the solution (Figure 6c), indicating that the stability of the *Zn-saldmpn* probe molecules was poor, and even minor changes in the external environment could significantly impact the molecular state.

In order to evaluate the anti-interference ability of the probe, we studied the fluorescence response of *Zn-3,5-Cl-saldmpn* and *Zn-3,5-Br-saldmpn* to common interferents, including common metal ions and anions. As shown in Figure 4e, Figure 6d and Appendix A, the fluorescence spectra of *Zn-3,5-Cl-saldmpn* and *Zn-3,5-Br-saldmpn* remained constant in the presence of these interferents. In contrast, when the pH changed to 5.0, the fluorescence intensity of *Zn-3,5-Cl-saldmpn* and *Zn-3,5-Br-saldmpn* decreased significantly. These results indicated that the halogen-modulated Schiff base probes were only sensitive to pH and were not affected by common interferents.

Reversibility is another important indicator of pH probes. Therefore, the fluorescence intensities of *Zn-3,5-Cl-saldmpn* and *Zn-3,5-Br-saldmpn* were measured under alternate acidic and alkaline environments. We adjusted the pH environment of the probe using two solutions with pH = 3.0 and pH = 11.0, respectively, and measured the fluorescence intensity (Appendix A).

The experimental results (Figure 7a) showed that *Zn-3,5-Cl-saldmpn* exhibited high fluorescence at pH 7.0, and the fluorescence was quenched after adding acidic solutions, but quickly recovered after adding basic solutions. The reversibility of the fluorescence intensity could be repeated more than 20 times (Figure 7c), and *Zn-3,5-Br-saldmpn* exhibited a similar phenomenon (Appendix A). However, due to the low signal-to-noise ratio and poor stability of *Zn-saldmpn*, it was unable to exhibit the reversible behavior of the non-halogenated analogs (Figure 7b and Appendix A). These experimental results indicated that the halogen-regulated Schiff base probes had excellent reversible response performance and could achieve the real-time and accurate monitoring of pH changes.

### 2.6. Colorimetric Sensing Test Strip and Smartphone Visual Detection Based on Zn-3,5-Cl-Saldmpn

By visually observing the fluorescence intensity of the probe test paper, it was found that as the pH value increased, the color of the fluorescence gradually changed from black to blue, and the fluorescence intensity was enhanced accordingly (Appendix A). Although the results can be analyzed by visual colorimetric comparison, this method relies on the human ability to discern colors, resulting in significant uncertainties in accuracy and sensitivity [53]. In order to explore quick and easy real-time colorimetric pH sensing via *Zn-3,5-Cl-saldmpn*, image studies based on mobile phone imaging systems were carried out since this type of device is portable and could greatly simplify the experimental procedure and reduce detection cost.

To achieve this, the values of RGB were extracted through a color analysis app on a mobile phone and plotted against the pH value. With increasing pH, the color gradually changed from black to blue, resulting in a significant increase in blue color, by which the pH value could be quantitatively calculated. Slope fitting was performed based on the blue channel (Appendix A), R^2^ was 0.995, and the fitting formula could be used to precisely calculate the pH value of an unknown solution via the imaging system from the blue channel value.

## 3. Materials and Methods

### 3.1. Materials and Instruments

All reagents and chemicals used in the experiments were of analytical reagent (AR) grade. The N′, N-bis(3-aminopropyl) methylamine (98%), 3,5-dichlorosalicylaldehyde (98%), 3,5-dibromosalicylaldehyde (98%), salicylaldehyde (97%), zinc chloride (Zn^2+^, Cl^−^, 99%), nickel sulfate (Ni^2+^, SO_4_^2−^, 98%), cupric acetate (Cu^2+^, Ac^−^, 99%), aluminum carbonate (Al^3+^, CO_3_^2−^, 99%), magnesium perchlorate (Mg^2+^, ClO_4_^−^, 99%), cobalt nitrate(Co^2+^, NO_3_^−^, 99%), sodium thiocyanate (Na^+^, SCN^−^, 98.5%), cerium nitrate (Ce^3+^, 99%), europium chloride (Eu^3+^, 99.9%), calcium fluoride (Ca^2+^, F^−^, 99.5%), potassium bromide (K^+^, Br^−^, 99%), manganese acetate (Mn^2+^, 99%), titanium tetrachloride (Ti^4+^, 99%), iron nitrate (Fe^3+^, 98.5%), strontium chloride (Sr^2+^, 99.5%), silver nitrate (Ag^+^, 99%), and chromium nitrate (Cr^3+^, 99%) were purchased from Aladdin Reagent Co., Ltd. (Shanghai, China). The ethanol (EtOH, 99.7%) was purchased from Tianjin Damao Company. Throughout the experimental process, ultrapure water was used; all reagents were used without further purification.

UV–vis absorption spectra were recorded by a SHIMADZU UV-2450 (Kyoto, Japan). Fourier transform infrared spectra were recorded at a scan range of 4000–400 cm^−1^ with a resolution of 4 cm^−1^, and conventional KBr detected the samples on a Thermo Scientific Nicolay 6700 DTSG-KBr detector (Thermal Sciences, Saint Louis, MO, USA). For the method of particle size measurement, a Mastersizer 3000 (UK) dynamic laser scattering (DLS) method was used to detect the particle size change of the molecules in solvents with different pHs. Fluorescence spectra, fluorescence lifetimes, and absolute PL quantum yields were measured by an Edinburgh FS5 fluorescence spectrometer (Livingston, UK). Single-crystal data for the reported compounds were collected at a low temperature (100 K) and by a Bruker APEX-II CCD diffractometer (Billerica, MA, USA) equipped with a CCD detector (Bruker) using graphite monochromatic MoKα radiation (λ = 0.71073 Å). For the measurements, ShelXT was invoked using the program Olex2, the structure was solved by the direct method, and the remaining atoms were solved from successive difference Fourier synthesis. The software calculated the positions of hydrogen atoms connected to salicylaldehyde and amine groups. All non-hydrogen atoms were anisotropically refined. All fluorescence photos were collected under a 365 nm UV lamp using a smartphone (Oppo Reno 8 Pro) in a dark box utilizing a ZF-1 UV lamp (Jiapeng Technology Co., Ltd., Shanghai, China). The pH values of all solutions were precisely measured with a Denver UB–7 digital pH meter.

### 3.2. pH Measurements

The pH measurements of *Zn-3,5-Cl-saldmpn*, *Zn-3,5-Br-saldmpn* and *Zn-saldmpn* were studied by using HCl or NaOH in an ethanol/water mixture (20:1, *v*/*v*). The ethanol solutions of HCl and NaOH were added to the solution of *Zn-3,5-Cl-saldmpn* (c = 400 µM), respectively. The fluorescence emission spectra of the solutions were measured at every pH unit (*Zn-3,5-Cl-saldmpn*: λex = 380 nm; *Zn-3,5-Br-saldmpn*: λex = 370 nm; *Zn-saldmpn*: λex = 355 nm).

### 3.3. Measurements of Fluorescence Lifetime

The preparation of the solution was exactly the same as for the pH measurements. The fluorescence lifetimes of *Zn-3,5-Cl-saldmpn*, *Zn-3,5-Br-saldmpn* and *Zn-saldmpn* at different pHs (5.0, 6.0, 7.0) were measured by Edinburgh FS5. Excitation at 380 nm was achieved using a picosecond pulsed diode laser (Edinburgh instruments, EPL-380) at a repetition rate of 100 Hz, the channel was 1024 (time/ch: 0.05 ns), the time range was 20 ns, the peak counts were 1000, and the monochromator bandwidth was 1.5 nm. The measured decay curve was fitted by Origin, the fitting function was ExpDecay, and the fitting order was 2.

### 3.4. Anti-Interference Experiment

First, the prepared *Zn-3,5-Cl-saldmpn* and *Zn-3,5-Br-saldmpn* solutions were diluted to 400 µM and shaken at room temperature for 10 s to ensure thorough mixing. Then, different kinds of interferents were added, mixed with *Zn-3,5-Cl-saldmpn* and *Zn-3,5-Br-saldmpn*, and the fluorescence spectrum was immediately measured (*Zn-3,5-Cl-saldmpn*: λex = 380 nm; *Zn-3,5-Br-saldmpn*: λex = 370 nm).

### 3.5. Stability and Reversibility

Ethanol/water solutions (20:1, *v*/*v*) (c = 400 µM) of *Zn-3,5-Cl-saldmpn* and *Zn-3,5-Br-saldmpn* were prepared. The pH value was regulated between acidic and alkaline using aqueous HCl (1 × 10^5^ µM) and NaOH (1 × 10^5^ µM) solutions. In addition, samples of the same concentration (c = 400 µM) of pH 5.0 and 7.0 were exposed to daylight, and their fluorescence intensity measurements were taken at regular intervals (*Zn-3,5-Cl-saldmpn*: λex = 380 nm; *Zn-3,5-Br-saldmpn*: λex = 370 nm; *Zn-saldmpn*: λex = 355 nm).

### 3.6. Preparation of Sensing Filter Paper

A circular filter paper with a radius of 4 cm was soaked in *Zn-3,5-Cl-saldmpn* solution with a concentration of 1 × 10^−2^ mmol/mL for one day to ensure that the solute could be absorbed on the surface and evenly dispersed in the pores of the filter paper, and then it was removed with tweezers, placed in a beaker, and dried in an oven at 105 °C for 20 min to obtain a fluorescence-sensing filter paper. The prepared filter paper was placed under a 365 nm ultraviolet lamp, dropped into ethanol with different pH values (5.0–7.0) solution (5 μL), and then visually observed in the dark.

### 3.7. Computational Methods

Previous studies have indicated that the B3LYP method is able to achieve satisfactory results in explaining the reaction mechanism and reactivity of Schiff base complexes [54]. Therefore, in this work the DFT method based on the standard 6-311 + G(d,p) basis set was employed to obtain and probe the relationship between molecular orbitals and bonding energies [49]. All calculations were carried out using Gaussian 16W [55].

## 4. Conclusions

In summary, we designed and synthesized two different halogenated Schiff base complexes (*Zn-3,5-Cl-saldmpn*, *Zn-3,5-Br-saldmpn*) and systematically compared the photophysical properties with non-halogenated analogous (*Zn-saldmpn*). According to the experimental results, the halogen-regulated Schiff base fluorescence probe not only showed excellent stability, selectivity, and reversibility, but also exhibited a high signal-to-noise ratio when the pH dropped from 7.0 to 5.0. The signal-to-noise ratios of *Zn-3,5-Cl-saldmpn* and *Zn-3,5-Br-saldmpn* were 12.2 and 6.7, respectively, while the signal-to-noise ratio of the non-halogenated Schiff base complex was only 1.7. Furthermore, theoretical studies have shown that the introduction of halogen atoms can not only control the surface charge distribution of the benzene ring, improve molecular conjugation, and increase intermolecular interactions, but also form more non-radiative pathways. Moreover, the introduction of halogen atoms can also change the packing arrangement of the probes, increasing the interaction energy between adjacent molecules and increasing the degree of the aggregation state. Theoretical studies have also been further supported by experimental data, including dynamic light scattering (DLS), scanning electron microscopy (SEM), and the Tyndall Effect. Our research, to some extent, can provide new guidance for the design and synthesis of new probes and is expected to enhance the SNR of reported probes.

## Data Availability

Not applicable.

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
