# Peer review of "Supramolecular Dimer as High-Performance pH Probe: Study on the Fluorescence Properties of Halogenated Ligands in Rigid Schiff Base Complex"

_ijms, 2023, doi:10.3390/ijms24119480_

Round 1

Reviewer 1 Report

The text describes the development of two new pH sensors, namely Zn-3,5-Cl-saldmpn and Zn-3,5-Br-saldmpn, which are based on a halogenated Schiff ligand. These sensors exhibit linearity, high signal-to-noise ratio, and demonstrate a significant increase in fluorescence emission and chromatic shift as the pH increases from 5.0 to 7.0. The sensors also show excellent stability and reversibility, retaining over 95% of their initial signal amplitude after 20 operational cycles.

What is the main aim of this study?

What are the possible applications of these pH sensors?

Lines 110, 122 and 134: showed a strong peak absorption peak = showed a strong absorption peak; Line 243: Further more= Furthermore: Line 242: can significantly = can be significantly: Line 372: halogenated ligand = halogenated ligands: line 462: and reducing detection = and reduce detection: line 528: observe the filter = observed the filter.

The text describes the development of two new pH sensors, namely Zn-3,5-Cl-saldmpn and Zn-3,5-Br-saldmpn, which are based on a halogenated Schiff ligand. These sensors exhibit linearity, high signal-to-noise ratio, and demonstrate a significant increase in fluorescence emission and chromatic shift as the pH increases from 5.0 to 7.0. The sensors also show excellent stability and reversibility, retaining over 95% of their initial signal amplitude after 20 operational cycles.

What is the main aim of this study?

What are the possible applications of these pH sensors?

Lines 110, 122 and 134: showed a strong peak absorption peak = showed a strong absorption peak; Line 243: Further more= Furthermore: Line 242: can significantly = can be significantly: Line 372: halogenated ligand = halogenated ligands: line 462: and reducing detection = and reduce detection: line 528: observe the filter = observed the filter.

Reviewer 2 Report

The article "Supermolecular Dimer as High-Performance pH Probe: Study on the Fluorescence Properties of Halogenated Ligands in Rigid Schiff Base Complex" describes a simple fluorescent method for determining pH in an ethanol/water mixture. The work is of good quality but needs improvement based on the following comments:

Major points:

1. The accuracy of pH measurement needs to be addressed since the authors claim to have developed pH probes. Will the pH value in water and the ethanol/water mixture be the same or different? Which electrodes were used for pH measurement, and how were they calibrated for the ethanol/water mixture (20:1 v:v)? This information should be added to the experimental section.

2. To confirm the structure of the obtained probes, NMR data or crystallographic data should be provided for Zn-3,5-Br-saldmpn and Zn-saldmpn. IR spectroscopy data are insufficient for this purpose. Additional crystallographic data are available only for Zn-3,5-Cl-saldmpn (CCDC 2254822). Unfortunately, data (CCDC 2254823) are not accessible.

3. The authors mention that ligand-to-metal charge transfer (LMCT) is observed in the complexes (page 3), but the theoretical calculations indicate that the zinc atom does not participate in constructing the HOMO and LUMO orbitals. An explanation is required.

Minor points:

1. The authors should explain why some measurements were taken on a 10:1 (v:v) EtOH/water mixture and others on a 20:1 (v:v) mixture.

2. The materials and methods section should include details on how time-resolved fluorescence spectroscopy was performed, such as the number of exponents used to describe the decay curves and the χ2 description of the experimental data.

3. In the materials and methods section, it is written that the basic set 6-311+G(d,p) was used, but in the discussion of the results, 6-311g(d,p) is mentioned. The authors should clarify which one was used.

Round 2

Reviewer 2 Report

Dear Authors

I am satisfied with your answers to the questions. For further pH measurements in aqueous-organic media, I recommend reading papers 10.1007/s00216-002-1455-z, 10.3390/s21113935